# The Antiviral Potential of *Perilla frutescens*: Advances and Perspectives

**DOI:** 10.3390/molecules29143328

**Published:** 2024-07-15

**Authors:** Jing Chen, Yi Zhao, Jie Cheng, Haoran Wang, Shu Pan, Yuwei Liu

**Affiliations:** 1Department of Bioinformatics and Intelligent Diagnosis, School of Medicine, Jiangsu University, Zhenjiang 212003, China; 2212313062@stmail.ujs.edu.cn (J.C.); 2212113042@stmail.ujs.edu.cn (Y.Z.); chengjie@ujs.edu.cn (J.C.); 1000005528@ujs.edu.cn (H.W.); 2Computer Science School, Jiangsu University of Science and Technology, Zhenjiang 212003, China; jsjxy_ps@just.edu.cn

**Keywords:** *Perilla frutescens*, antiviral, natural drug, active compounds, oral bioavailability (OB)

## Abstract

Viruses pose a significant threat to human health, causing widespread diseases and impacting the global economy. *Perilla frutescens*, a traditional medicine and food homologous plant, is well known for its antiviral properties. This systematic review examines the antiviral potential of *Perilla frutescens*, including its antiviral activity, chemical structure and pharmacological parameters. Utilizing bioinformatics analysis, we revealed the correlation between *Perilla frutescens* and antiviral activity, identified overlaps between *Perilla frutescens* target genes and virus-related genes, and explored related signaling pathways. Moreover, a classified summary of the active components of *Perilla frutescens*, focusing on compounds associated with antiviral activity, provides important clues for optimizing the antiviral drug development of *Perilla frutescens*. Our findings indicate that *Perilla frutescens* showed a strong antiviral effect, and its active ingredients can effectively inhibit the replication and spread of a variety of viruses in this review. The antiviral mechanisms of *Perilla frutescens* may involve several pathways, including enhanced immune function, modulation of inflammatory responses, and inhibition of key enzyme activities such as viral replicase. These results underscore the potential antiviral application of *Perilla frutescens* as a natural plant and provide important implications for the development of new antiviral drugs.

## 1. Introduction

The role of viral infection in human diseases is significant, and ensuring the prevention of viral infection is a crucial aspect in safeguarding public health. Certain infectious diseases demonstrate extensive spread and high infectivity, profoundly impacting the global economy and politics. Furthermore, some viruses can induce chronic infectious conditions such as human immunodeficiency virus (HIV) [1,2], hepatitis B virus (HBV) [3,4], and hepatitis C virus (HCV) [5,6]. These conditions progress gradually and chronically, leading to reduced labor capacity in patients and decreased life expectancy. Consequently, these viruses profoundly affect both the quality of life for patients as well as economic aspects. The association between certain tumors and viruses is well-established, such as Epstein Barr virus with nasopharyngeal carcinoma [7,8], human papillomavirus with cervical cancer [9,10,11], and human herpesvirus type 8 (HHV-8) with Kaposi’s sarcoma [12,13]. Viruses exhibit a high mutation rate and continuously generate new variants, posing a significant threat to human health.

The antiviral potential of numerous natural compounds has been demonstrated in various studies, revealing the ability of numerous plant extracts and secondary metabolites to effectively inhibit viral replication and transmission [14]. The mechanisms of antiviral action are diverse, encompassing interference with viral entry into host cells, inhibition of viral gene expression, disruption of viral assembly, and augmentation of the host immune response [15]. For instance, flavonoids primarily inhibit viral protease activity to prevent viral replication [16]. On the other hand, terpenoids mainly interfere with the fusion of viruses and host cell membranes to impede virus entry into host cells [17]. Additionally, certain polyphenolic compounds directly hinder the cytopathic effect [18]. These findings establish a crucial scientific foundation for the development of novel antiviral medications.

The annual herb *Perilla frutescens* (L.) Britt., belonging to the Labiatae family, exhibits medicinal and culinary properties in traditional Chinese medicine (TCM) [19]. Its dried stems, leaves, and seeds have been utilized as medicinal materials. *Perilla frutescens* has demonstrated pharmacological activities, including anti-fungal [20], antiviral [21], anti-cancer [22,23], hypoglycemic, and heart-protective effects [24,25]. In this paper, the Traditional Chinese Medicine Systems Pharmacology Database and Analysis Platform (TCMSP) (https://old.tcmsp-e.com/tcmsp.php, accessed on 23 July 2023) were used to sort all reported monomer components of *Perilla frutescens* by oral bioavailability (OB). The term “oral bioavailability” (OB) refers to the extent and rate at which a drug is absorbed into the systemic circulation. It serves as a crucial parameter for objectively assessing both the oral bioavailability and intrinsic quality of a drug, while also serving as a pivotal criterion for determining its potential as a therapeutic agent. A higher OB value indicates an increased likelihood of clinical development for the compound [26]. Then, the names of monomer components of *Perilla frutescens* exhibiting OB values exceeding 20% were searched in PubMed and Web of Science databases, using keywords such as “antiviral”. More than 200 recently published papers were reviewed and discussed. This paper focuses on reviewing the antiviral activities of these primary components and their derivatives. The antiviral mechanisms are described in terms of the chemical structure, pharmacological parameters, and bioavailability of these components against viruses.

This review classification summarizes the antiviral abilities of monomeric components of *Perilla frutescens* with OB greater than 20% against various viruses and their mechanism of action. Furthermore, exploring potential synergistic effects by combining these drugs could pave the way for developing more effective antiviral strategies. In summary, *Perilla frutescens* shows promising potential as an antiviral drug candidate, highlighting the need for further preclinical studies and clinical trials.

## 2. The Application of Bioinformatics Analysis to Explore the Correlation between *Perilla frutescens* and Antiviral Activity

The keyword “*Perilla frutescens*” was initially searched in the TCMSP firstly, resulting in the retrieval of 2481 target names. To facilitate analysis, the target proteins were transformed to standard gene symbols, ultimately yielding 397 gene symbols. Similarly, a total of 190 *Perilla frutescens* related target genes were obtained from the Encyclopedia of Traditional Chinese Medicine (ECTM, http://www.tcmip.cn/ETCM/, accessed on 24 October 2023), while an additional 484 *Perillae Folium* related target genes were identified in the Symptom Mapping (SymMap, https://www.Symmap.org/, accessed on 25 October 2023). After merging these obtained target genes and removing duplicates, a comprehensive set of 671 unique *Perilla frutescens*-related target genes was identified.

Furthermore, we employed the GeneCards (https://www.genecards.org/, accessed on 31 October 2023), a comprehensive resource for disease-related information [27], to extract target genes associated with herpes simplex virus (HSV), Severe Acute Respiratory Syndrome Coronavirus-2 (SARS-CoV-2), influenza virus, and human immunodeficiency virus (HIV). This enabled us to integrate these virus-associated target genes with the *Perilla*-related target genes for subsequent analysis.

The analysis revealed a significant overlap between 671 *Perilla* target genes and 2791 HSV-related target genes, 6899 SARS-CoV-2-related target genes, 3211 influenza virus-related target genes, and 9015 human immunodeficiency virus-related targets. This indicates that *Perilla frutescens* exhibits substantial antiviral potential, as depicted in Figure 1.

To further investigate the biological function of *Perilla frutescens*, we utilized the Database for Annotation, Visualization and Integrated Discovery (DAVID, https://david.ncifcrf.gov/, accessed on 5 November 2023) to analyze the comprehensive set of target genes associated with *Perilla frutescens,* and conducted separate analyses to assess Gene Ontology (GO) enrichment and Kyoto Encyclopedia of Genes and Genomes (KEGG) pathway enrichment of these target genes. The GO terms in Biological Process (BP), Molecular Function (MF), and Cellular Component (CC) categories were ranked based on gene ratio applying Benjamini–Hochberg correction at a significance level of *p* < 0.05.

Through GO enrichment analysis on the DAVID platform, the predicted enrichments of *Perilla frutescens* targets primarily involve the regulation of signal transduction, positive transcriptional regulation from RNA polymerase II promoter, negative regulation of apoptotic process, and inflammatory response, as illustrated in the top 10 biological processes (BP) shown in Figure 2A. Furthermore, the GO enrichment analysis for overlapping genes between *Perilla frutescens* and diseases associated with viral infections aligns with the GO results for total targets of *Perilla frutescens* as depicted in Figure 2B. This suggests that these molecular functions may play a crucial role in enhancing *Perilla frutescens’* antiviral capability.

The KEGG pathway enrichment analysis identified the top 20 enriched terms in *Perilla frutescens* using the DAVID database, revealing associations with pathways involved in cancer and viral infection. Furthermore, comparison of overlapping genes between *Perilla frutescens* and diseases associated with viral infections showed that 90% of the top 20 enriched terms aligned with the KEGG results obtained for total drug predicted targets. These findings underscored the importance of further investigation into these signaling pathways to advance research on antiviral treatments, as depicted in Figure 3.

The integration of bioinformatics analysis suggests that *Perilla frutescens*, might play a pivotal role in antiviral processes by regulating diverse biological mechanisms and exerting antiviral effects through the aforementioned molecular mechanisms.

## 3. The Antiviral Properties of Monomer Components Derived from *Perilla*

From the TCMSP database, we retrieved the monomeric constituents of *Perilla frutescens*, ranked their OB, and selected the top 20% with higher OB (%) values for the literature review using the PubMed database. Based on compound classification, our objective was to summarize the reported active components associated with antiviral activity.

### 3.1. Phenols

Phenols and phenolic ethers are significant scaffolds found commonly both in nature and among approved small-molecule pharmaceuticals [28]. These natural phenolic compounds are widespread in plants, where they serve protective roles against ultraviolet radiation and other forms of harm [29]. Phenolic compounds possess significant pharmacological and nutritional characteristics, including anti-inflammatory [30], antibacterial [31], antiviral [32], and antioxidant [33,34] properties among others. As their value continues to be recognized by the scientific community, they may emerge as important micronutrients. Phenolic compounds are recurrent and significant moieties in nature, playing a crucial role as authorized small molecule drugs [28]. Furthermore, they exhibit inhibitory effects on viral replicase such as HIV reverse transcriptase and RNA polymerase of influenza virus [35]. The antiviral activity of numerous polyphenols, including resveratrol, curcumin, epigallocatechin gallate, indigo, aloe-emodine, quinomethyl triterpenoids, quercetin or gallates has been identified through computer simulations and in vitro studies involving cell-free polyphenol–protein interactions and cell-based viral infection models [36]. Some of these compounds demonstrate promising potential to emerge as dominant agents for COVID-19 therapeutics.

#### 3.1.1. Thymol

The natural phenolic monoterpenoid thymol (2-isopropyl-5-methylphenol) is primarily extracted from Thymus species [37]. In the field of traditional Chinese medicine, thymol has been utilized for an extended period as an expectorant [38], anti-inflammatory [39], antiviral [40], and antibacterial [41,42] agent specifically targeting upper respiratory system diseases.

Thymol exhibited antiviral activity against influenza virus, herpes simplex virus type 1 (HSV-1) [43], and HIV [44]. It demonstrated the ability to inhibit replication of IAV and suppress inflammatory mediators, thereby restraining pneumonia development. Additionally, it reduced interleukin (IL)-4 and IFN (interferon)-γ levels in serum while enhancing antioxidant activity in lung tissue [45]. These findings suggest that thymol could be a potential drug candidate for treating influenza infections in mice.

Thymol exhibited antiviral activity by effectively binding to the receptor of SARS-CoV-2 spike glycoprotein S1 [46]. Transmembrane protease serine 2 (TMPRSS2) proteins facilitated virus internalization by cleaving the spike protein of SARS-CoV-2 [47]. Stable binding of thymol with TMPRSS2 induces subtle spatial rearrangements in catalytic triad residues, making it an excellent inhibitor against SARS-CoV-2.

Additionally, thymol has been documented to exhibit antiviral activity against various viruses, including norovirus surrogates [48], feline calicivirus (FCV) [48], murine norovirus (MNV) [48], bovine viral diarrhea virus (BVDV) [49], tomato leaf curl New Delhi virus [50], and Cyprinid herpesvirus 3 (CyHV-3) [51].

#### 3.1.2. Eugenol

Eugenol, a phenolic aromatic compound primarily derived from clove oil [52], plays a crucial role in the innate immune response against viruses [53].

Eugenol has antiviral properties against IAV [54] and HSV [55], inhibiting viral replication. It can also enhance the effectiveness of acyclovir in inhibiting HSV replication. Additionally, topical application of Eugenol delays the development of herpes virus keratitis in mice models [56]. Meanwhile, Eugenol exhibited potential binding characteristics with the main proteinase of SARS-CoV-2 [57]. Network analysis revealed that Eugenol interacted with host proteins ACE2, DPP4, COMT, TUBGCP3, CENPF, BRD2 and HMOX1 [58], thereby playing an antiviral role in virus entry, viral replication, host immune response and antioxidant activity. Moreover, Eugenol effectively mitigated intestinal heat necrosis caused by transmissible gastroenteritis virus (TGEV) by suppressing the activation of NLRP3 inflammasome. This effect may be mediated through modulation of intracellular ROS levels. These findings suggest Eugenol as a promising strategy for preventing TGEV infection [53]. Additionally, the results of molecular docking research have also shown Eugenol’s inhibitory effects on Dengue virus by interacting with the NS1 and NS5 proteins [59]. Moreover, Eugenol has been documented to exhibit antiviral activity against tomato yellow leaf curl virus (TYLCV) [60], tomato yellow leaf curl Thailand virus (TYLCTHV) [61], human norovirus, tobacco mosaic virus (TMV) [62], HCV [63], and Feline Calicivirus (FCV) [64].

#### 3.1.3. Protocatechualdehyde

The compound protocatechualdehyde demonstrated significant anti-HBV activity. In vitro, protocatechualdehyde effectively suppressed the secretion of hepatitis B e antigen (HBeAg) and hepatitis B surface antigen (HBsAg) and reduced the release of HBV DNA in the HepG2 2.2.15 cell line. In vivo, protocatechualdehyde diminished viremia in DHBV-infected ducks. As a novel therapeutic agent against HBV, protocatechualdehyde exhibited potential as an efficacious treatment for HBV infections [65]. The combination of angiotensin-converting enzyme 2 (ACE2) and the spike protein of SARS-CoV-2 enables viral entry into host cells by crossing the cell membrane. Transmembrane protease serine 2 (TMPRSS2) modifies SARS-CoV-2 to facilitate cellular access, making ACE2 and TMPRSS2 crucial targets for preventing virus infection. In cell lines and mouse models infected with SARS-CoV-2, protocatechualdehyde demonstrated potential activity in reducing the expression of ACE2 and TMPRSS2, suggesting its efficacy in preventing SARS-CoV-2 infection [66].

#### 3.1.4. Methyl Caffeate

Methyl caffeate (MC) significantly inhibited the replication of HIV in peripheral blood mononuclear cells (PBMCs) without causing noticeable cytotoxicity. In mice infected with HIV, different doses of MC treatment led to varying degrees of increased expression of IL-2, IL-4, interferon-gamma (IFN-g), and soluble Fas. However, the expression of granulocyte-macrophage colony-stimulating factor (GM-CSF) remains unaffected by MC. These findings suggest that MC has potential as a chemotherapy agent for anti-HIV infection and cytokine regulation, warranting further investigation [67].

Table 1 presented the antiviral activity of the main phenols found in *Perilla frutescens*. The evaluation of drug similarity is crucial in the production and upgrading of drug entities [68]. We first predicted the physicochemical properties of main phenols according to Lipinski’s rule of five (Ro5) using Molinspiration cheminformatics (https://molinspiration.com/, accessed on 8 November 2023). The criteria for the Rule of Five (Ro5) are as follows: LogP should be less than or equal to 5, molecular weight (MW) should be less than or equal to 500 Da, the number of hydrogen bond acceptors (n-ON) should be less than or equal to 10, and the number of violations in terms of hydrogen bond donors (n-OHNH) should be less than or equal to 5. The compounds that conform to the Rule of Five (Ro5) exhibit improved pharmacokinetic properties, enhanced bioavailability in biological metabolism, and therefore possess a higher likelihood of being developed into oral medications [69]. The topological polar surface area (TPSA), which utilizes functional group contributions derived from a comprehensive database of structures, serves as a convenient metric for quantifying the extent of polar surface area [70] and a TPSA value ≤ 140 Å represents good oral bioavailability [68]. The results demonstrated that the main phenols fulfilled the criteria of Rule of Five (Ro5) and exhibited a TPSA value ≤ 140 Å, as presented in Table 2. The biological activity analysis of compounds, conducted using Molinspiration cheminformatics, encompassed G-protein coupled receptor (GPCR) ligands, ion channel modulators, kinase inhibitors, nuclear receptor ligands, protease inhibitors, and enzyme inhibitors. A bioactivity score >0 indicated promising activity; a score between −0.50 and 0.00 represented moderate activity; while a score ≤ −0.50 indicated no activity [68,71]. These findings suggest that phenolic compounds possess moderate affinity as ion channel modulators. Physicochemical properties are shown in Table 2 and the bioavailabity of compounds is seen in Table 3.

### 3.2. Terpenoids

Terpenoids, also referred to as isoprenoids, constitute a class of secondary metabolites found in plants characterized by diverse chemical structures. They are widely recognized as a vast reservoir of bioactive compounds in nature and hold significant industrial and pharmaceutical value [72,73]. Terpenoids have been recognized for their multifaceted pharmacological effects including anti-cancer [74,75], anti-inflammatory [76,77], antioxidant [76,78], analgesic [79], antibacterial [80,81], antifungal [81,82], hepatoprotective [79,83], antiviral [84,85] and antiparasitic activities [86,87].

#### 3.2.1. Perillyl Alcohol

Perillyl alcohol is classified as a type of terpenoid compound [88]. It has been scientifically proven to possess potent anti-tumor [89,90], antiviral [91], anti-inflammatory [92], and antibacterial [93] properties. Protein separation analysis has demonstrated that perillyl alcohol exhibited strong antiviral activity [94], particularly against HSV-1. The replication of viral genomes was not inhibited by Perillyl alcohol; instead, it effectively suppressed the release of infectious viral particles in Vero cells. This suggested that Perillyl alcohol exerted its effects during the late maturation stage of HSV-1 and holds great potential for clinical anti-HSV-1 therapy to prevent intermittent reactivation and progressive gray matter loss [91]. In vitro studies have shown that treatment with perillyl alcohol could reduce respiratory syncytial virus (RSV) infection by inhibiting host protein prenylation, including the prenylation of Rho GTPases [95].

#### 3.2.2. Germacron

Germacron, a natural terpenoid, has garnered significant attention due to its diverse pharmacological properties including anticancer [74,96], antiviral [97,98], antioxidant [99] and antibacterial effects [96].

The biological activities of germacron are attributed to its ketone and non-conjugated double bonds [100]. Studies have demonstrated that germacron activated transcription of interferon genes and protects peripheral cells from IAV infection [100]. Moreover, it exhibited dose-dependent antiviral activity against H1N1 and H3N2 influenza A viruses (IAV) as well as influenza B viruses (IBV) by reducing viral protein expression, RNA synthesis and production of infectious progeny viruses in vitro [101]. It also reduced H1N1-induced lung injury and viral load in serum and whole blood cells while decreasing the expression of antiviral proteins [102]. Furthermore, combination therapy with germacone and oseltamivir showed an additive effect on inhibiting influenza virus infection both in vitro and in vivo suggesting that germacone could be a potential therapeutic agent for treating influenza virus infections alone or in combination with other drugs [101]. Additionally, germacron protected cells from Porcine parvovirus (PPV) infection by suppressing viral mRNA/protein synthesis while inhibiting PRRV replication remarkably without blocking PRRS binding/entry [97,103]. In addition, germacron was found to have potential as a therapeutic agent for treating Porcine Reproductive and Respiratory Syndrome Virus (PRRSV) infection [97]. In vitro studies demonstrated that germacron effectively inhibited the growth of Feline calicivirus (FCV) strain F9 and exhibited strong antiviral effects against FCV primarily during the early stages of its life cycle [104], with efficacy dependent on drug concentration. Meanwhile, germacron treatment significantly suppressed the replication rates of reference strains 2280 and Bolin, as well as WZ-1 and HRB-SS strains isolated from China [97]. Additionally, germacron dose-dependently inhibited Pseudo rabies virus (PRV) replication in vitro and displayed antiviral activity against PRV during the initial phases of viral replication. Importantly, it did not directly kill the virus nor affect the expression of PRV receptor proteins nectin-1, nectin-2, or CD155, and the possible antiviral mechanism was affecting the cellular antiviral mechanisms [105].

#### 3.2.3. Patchouli Alcohol

The tricyclic sesquiterpene patchouli alcohol (PA) has been reported to possess a wide range of health-promoting activities, including antiviral effects against influenza [17,106], antidepressant properties [107], tissue-protective effects against injury [108,109], vascular relaxation abilities [110], lung and brain protective actions [111], anti-ulcer and anti-colitis activities [112], potent anti-inflammatory effects [113,114], potential anticancer properties as well as protective effects against metabolic diseases [115,116].

The compound PA exhibited potent antiviral activity against IAV both in vitro and in vivo. It demonstrated dose-dependent inhibition of influenza virus A/PR/8/34 (H1N1), while showing no activity against influenza virus A/54/89 (H3N2). However, it displayed weak antiviral activity against influenza virus B/Ibaraki/2/85 [117]. Notably, PA effectively suppressed viral replication during the early stages of IAV infection [17], suggesting its potential as a membrane fusion inhibitor for treating IAV infections. Furthermore, PA specifically hindered the expression of viral proteins hemagglutinin (HA) and nuclear protein (NP). In hemagglutinin inhibition (HAI) and hemolysis inhibition experiments, PA was able to block HA2-mediated membrane fusion at low pH levels with a lower binding energy to HA2 compared to HA1 [17]. Mechanistically, PA targeted the PI3K/Akt and ERK/MAPK signaling pathways within viral particles and cells to inhibit IAV infection. Intranasal administration of PA significantly improved survival rates in mice infected with IAV by reducing pneumonia symptoms, highlighting its potential as an effective antiviral drug against IAV infection [106]. Additionally, treatment with PA inhibited cytokine expression and RLH pathway mRNA levels associated with H1N1 influenza virus infection in vitro [118]. Moreover, through interference with neuraminidase function and cleavage of α-glycosidic bonds between sialic acid and glycoconjugates, PA also exhibited anti-influenza A (H2N2) virus activity [119]. Lastly, molecular docking studies combined with molecular dynamics simulations revealed that PA could serve as an inhibitor for SARS-CoV-2 enzymes including 3CL_pro_, PL_pro_, and NSP15 [120].

#### 3.2.4. Menthol

Peppermint essential oil (PEO), containing menthol as its primary component, exhibited anti-inflammatory [121,122], antibacterial [123], antiviral [124], and antioxidant properties [122]. Menthol activated transient receptor potential cation channel subfamily M member 8 (TRPM8), inducing a sensation of “cold” that reduced Coxsackievirus B infection and mitigated mitochondrial fission during infection. Additionally, menthol stabilized the level of mitochondrial antiviral signaling (MAVS) protein which is associated with mitochondrial dynamics [124]. It also influenced the proliferation of herpes simplex virus type 1 (HSV-1) and pseudorabies virus (PrV), without exhibiting cytotoxic effects at the tested concentrations [125].

The antiviral activity of main terpenoids in *Perilla frutescens*, with an OB value exceeding 20%, is presented in Table 4. Their physicochemical properties are listed in Table 5, while the bioactivity scores are displayed in Table 6. The results showed that main terpenoids met the criteria of Ro5 and the value of TPSA ≤ 140 Å and represented promising or moderate ion channel modulator and enzyme inhibitor affinity.

### 3.3. Flavonoids

The flavonoids, a prominent group of phytochemicals present in various plant species, have demonstrated significant antiviral activity against influenza virus and other RNA viruses [16,126]. They effectively impede viral replication and infectivity, modulate the host’s response to viral infection, and hold potential as promising candidates for antiviral therapeutics.

#### 3.3.1. Luteolin

Luteolin is a 3′, 4′, 5, 7-tetrahydroxyflavonoid that is widely present in various plants, fruits, and vegetables [127]. Evidence has shown its antiviral [128], anti-inflammatory [129,130], and immune regulatory functions [131].

Luteolin has been reported to inhibit multiple targets involved in SARS-CoV-2 replication including Papain-like protease [132,133,134], Coronavirus main proteinase [133,134,135], RNA nucleoside triphosphatases (NTPase)/helicases and Angiotensin-Converting Enzyme 2 [134]. In patients with long-term COVID-19 disease, luteolin could enhance GABAB-ergic activity and cortical plasticity while also showing potential for treating COVID-19/asthma comorbidity [136]. By regulating the NF-κB/STAT3/ATF6 signaling pathway, luteolin inhibited the replication of African swine fever virus in a dose-dependent manner [137]. It activated the cGAS-STING signaling pathway to produce IFNs with antiviral effects against herpes simplex virus 1 infection [138]. Luteolin exhibited antiviral activity against chikungunya virus (CHIKV) without causing cytotoxicity [139]. While it showed no antiviral activity during viral binding and entry stages of Japanese encephalitis virus (JEV) infection in vitro, it demonstrated cell killing activity against extracellular JEV particles [140]. Moreover, in vitro and in vivo studies have revealed that luteolin distinctly inhibited Pseudorabies virus (PRV) replication as well as Feline infectious peritonitis virus at different stages of infection onset [128,141]. Luteolin effectively targeted the post-attachment stage of EV71 and CA16 infections by suppressing viral RNA replication [142]. In picornavirus life cycle where 3C protease plays an essential role, luteolin exhibited good binding affinity with foot-and-mouth-disease virus (FMDV) 3C_Pro_ thereby hindering FMDV life cycle through inhibition of its enzymatic activity. Hence, luteolin holds great potential as an antiviral agent not only against FMDV but also other picornaviruses [143].

#### 3.3.2. Apigenin

Apigenin, a flavonoid with low toxicity [144,145], exhibits diverse beneficial biological activities including anti-tumor [146,147], antioxidant [148,149], anti-inflammatory [150], and antiviral effects [126].

In vitro and in vivo studies have demonstrated the potent antiviral activity of apigenin against buffalopox virus (BPXV) [151]. Mechanistically, apigenin not only directly inhibited viral polymerase activity but also suppressed viral protein translation. Moreover, apigenin prevented cell death induced by IAV infection and reduces viral neuraminidase (NA) activity to impede viral replication. This inhibition is attributed to the disruption of heat shock protein 90α (Hsp90α) and retinoic acid-inducible gene-I (RIG-I) interaction by apigenin as well as the promotion of ubiquitin-mediated degradation of RIG-I to attenuate the RIG-I signaling pathway [144]. Furthermore, apigenin exerted its antiviral effect against foot-and-mouth disease virus (FMDV) through interference with FMDV translational activity driven by internal ribosomal entry site rather than direct extracellular virocidal action [152]. Additionally, apigenin inhibited the initiation of Epstein–Barr virus (EBV) lytic cycle via suppression of immediate early gene Zta and Rta promoter activation to prevent EBV reactivation [153]. Notably, previous reports have highlighted distinct antiviral properties of apigenin against various viruses such as HSV [154], EV71 [155,156], HCV [157,158], dengue virus (DENV) [159,160], and severe acute respiratory syndrome coronavirus (SARS-CoV) [161,162].

The antiviral activity of the main flavonoids in *Perilla frutescens* is presented in Table 7, while the physicochemical properties of these flavonoids are displayed in Table 8. Additionally, the bioactivity scores of the main flavonoids in *Perilla frutescens* can be found in Table 9.

### 3.4. Sterols

Sterols, which are isoprenoid derivatives, serve as integral components of biological membranes [163]. They have been widely identified in diverse marine and terrestrial sources (such as plants, animals, and microorganisms), making them a prevalent category of natural products. Moreover, sterols encompass numerous subclasses that exhibit a wide range of biological activities. Notably, sterols demonstrate potential antiviral effects against various viruses including herpes simplex virus (HSV) [154], and HBV [158]. The steroid class comprises 25 chemical subclasses with approximately 11,825 previously reported compounds [164].

#### β-Sitosterol

β-sitosterol has shown potential as an antiviral compound against HIV, HBV, IAV, SARS-CoV-2, WSSV, HSV-2, and TMV through various mechanisms such as immunomodulation and inhibition of viral replication.

β-sitosterol has been reported to effectively inhibit the pyrolysis process of the 3C-like protease (3CL_pro_) of SARS-coronavirus and closely interact with the main protease (M(pro)) of SARS-CoV-2 [165]. It showed potential as an herbal candidate with antiviral activity against SARS-CoV-2, making it valuable for future drug development against coronavirus diseases [166]. Additionally, β-sitosterol acted as a potent inhibitor of white spot syndrome virus (WSSV), significantly reducing viral load and viral gene transcription levels while improving survival in crayfish infected with WSSV [167]. Moreover, it exhibited antiviral activity against herpes simplex virus Type 2 (HSV-2) by directly inactivating viral particles and demonstrates efficacy against tobacco mosaic virus (TMV) [168]. Furthermore, β-sitosterol displayed great antiviral potential against avian and human IAVs in vitro [169].

Molecular docking studies have shown that β-sitosterol exerted its antiviral activity against IAV by binding to hemagglutinin protein, inhibiting viral replication through interference with viral neuraminidase and IAV M2 protein [169]. In addition, both in vivo and in vitro studies have demonstrated that β-sitosterol possesses anti-HIV activity through various immunomodulatory mechanisms such as stabilizing CD4+ T lymphocyte counts and significantly reducing interleukin-6 levels [170]. It also exhibited remarkable antioxidant properties along with being an anti-HBV agent [170,171], anti-Dengue virus 2 compound [172], and anti-IAV substance [173]. Bioinformatics analysis revealed that β-sitosterol participates in regulating the TRAF6/MAPK14 axis for its anti-influenza activity while displaying inhibitory effects on IAV nucleoprotein, polymerase, PBP2A, and DNA gyrase B [174,175]. Furthermore, it might exert its antiviral effects by inhibiting the fusion process to impede dengue virus-2 entry [172].

The antiviral activity of main sterols in *Perilla frutescens* were showed in Table 10. Physicochemical properties of main sterols in *Perilla frutescens* were showed in Table 11. Bioactivity scores of main sterols in *Perilla frutescens* were showed in Table 12. The results showed that β-sitosterol met the criteria of Ro5 and the value of TPSA ≤ 140 Å. Furthermore, β-sitosterol exhibited promising GPCR ligand, ion channel modulator, nuclear receptor ligand, protease inhibitor and enzyme inhibitor affinity.

### 3.5. Aldehydes

Aldehydes constitute a class of organic compounds with significant activity, naturally present in various food sources and ingredients, rendering them highly intriguing in both academic and industrial contexts. Aldehydes and their derivatives exhibit diverse effects, including antimicrobial [176], antioxidant [177], anti-inflammatory [178,179], and immunomodulatory properties [176]. The versatile applications of aldehyde products span across industries such as cosmetics and pharmaceuticals, making them promising targets for novel biological drugs [176,180].

#### Cinnamaldehyde

Cinnamaldehyde (CA) serves as a naturally occurring active ingredient well tolerated by both humans and animals. The safety of CA has been confirmed by the US Food and Drug Administration (FDA) and European Commission, who recommend a daily intake of 1.25 mg/kg [176]. Additionally, CA has been reported to possess numerous health benefits such as its application in the treatment of gastritis [181], dyspepsia [182], and blood circulation disorders [183].

The drug has been reported to interact with ACE 2, DPP4, COMT, TUBGCP3, CENPF, BRD 2 and HMOX 1 host proteins involved in antiviral mechanisms such as viral entry, replication, immune response and antioxidant activity against SARS-CoV-2 based on network pharmacology, virtual screening, molecular docking and molecular dynamics methods [58]. Meanwhile, it has been reported to possess anti-inflammatory effects and alleviate lung inflammation caused by SARS-CoV-2 infection [184].

Moreover, a combination of plant essential oils (PEO) from CA and glycerol monolaurate (GML) demonstrated significant inhibitory effects on infectious bronchitis virus (IBV), potentially through inhibition of viral proliferation and promotion of immune function. This suggested that PEO could be a promising novel anti-IBV drug for inhibiting IBV infection [185]. Furthermore, CA treatment inhibited the replication of influenza A/PR/8 virus in Madin–Darby canine kidney cells during the growth cycle without cytotoxicity in a dose-dependent manner. Direct application of CA in the airways significantly rescued fatal influenza virus-induced pneumonia and reduced lung virus production in infected mice [186].

Additionally, CA exhibited remarkable therapeutic effects against coxsackie virus B3 (CVB3)-induced viral myocarditis (VMC) [187]. Intraperitoneal injection of CA increased survival rates and decreased myocardial virus titers in VMC mice. Mechanistically, while cinnamic acid metabolites contributed to its antiviral activity against VMC by directly alleviating inflammatory reactions through inhibition of TLR4-NF-kappaB signal transduction pathways, brominated cassia bark aldehyde synthesized using CA as the lead compound not only enhanced antiviral activity against VMC but also exerted strong anti-inflammatory effects on b type cardiomyocytes [188].

The antiviral activity of aldehydes with an OB greater than 20% in *Perilla frutescens* is presented in Table 13, while the physicochemical properties of the main aldehydes in *Perilla frutescens* are displayed in Table 14. Additionally, the bioactivity scores of these main aldehydes are shown in Table 15. CA met the criteria of Ro5 and the value of TPSA ≤ 140 Å. Furthermore, CA exhibited moderate ion channel modulator and enzyme inhibitor affinity.

### 3.6. Others

In addition to the aforementioned compounds with a definite chemical classification, *Perilla frutescens* contains numerous compounds without a definitive chemical classification that have reported antiviral activity.

#### 3.6.1. Osthol

Osthol exhibited antiviral activity against tobacco mosaic virus (TMV) by directly targeting viral particles, making it a potential biological agent for controlling plant viruses using the half-leaf method [189]. Furthermore, osthol alone demonstrated antiviral activity against Porcine circovirus type 2 (PCV2), while its combination with Matrine resulted in better anti-PCV2 effects than either compound alone. The combined treatment of Matrine and osthol directly suppressed PCV2 Cap protein expression through the PERK pathway activated by endoplasmic reticulum GRP78, thereby inhibiting PCV2 Cap protein-induced PERK apoptosis and alleviating pathological changes such as interstitial pneumonia, splenic lymphocyte loss, macrophage infiltration, and eosinophil infiltration caused by PCV2 [190,191].

#### 3.6.2. Piperitenone

The compound piperitenone oxide exhibited antiviral activity by disrupting the later stages of the HSV-1 life cycle. Infection with HSV-1 resulted in depletion of the key antioxidant glutaglyanin within host cells, which was restored by PEO treatment. These findings suggested that this compound had the potential to interfere with redox-sensitive cellular pathways exploited for viral replication [192].

#### 3.6.3. Pulegone

Pulegone ((*R*)-5-Methyl-2-(1-methylethylidene) cyclohexanone), a pharmacologically active natural monoterpene ketone, exhibited antiviral activity against herpes simplex virus type 1 (HSV-1) and pseudorabies virus (PrV) [193], while demonstrating no cytotoxicity. Computational evaluation revealed that pulegone acted as a potent inhibitor of the SARS-CoV-2 spike protein, exerting antiviral effects [46]. Therapeutic administration of pulegone displayed antiviral activity against influenza virus; however, it did not exhibit significant preventive effects. The mechanism underlying its antiviral properties was associated with the regulation of IFN-alpha, IFN-beta, and IL-2 [46].

The antiviral activity of additional compounds found in *Perilla frutescens* is presented in Table 16. Physicochemical properties of these compounds are displayed in Table 17, while their bioactivity scores can be found in Table 18. The results showed that these compounds met the criteria of Ro5 and the value of TPSA ≤ 140 Å and exhibited promising and enzyme inhibitor affinity.

## 4. Discussion

Viruses can give rise to a range of diseases, including COVID-19 [194], hepatitis B [195], AIDS [196], influenza [197], and others. Certain viral infections have the potential to cause local and even global disruptions, posing substantial risks to public health.

The annual herb *Perilla frutescens* (L.) Britt., belonging to the Labiatae family, with a long-standing history in China, possesses an abundance of medicinal benefits [198]. Among numerous traditional Chinese medicine prescriptions, *Perilla frutescens* stands out due to its distinct antiviral efficacy and garners high comments from physicians throughout various dynasties. In the realm of Chinese traditional medicine, *Perilla frutescens* is often combined with other herbal materials to treat medical conditions such as colds, coughs, asthma, and other viral diseases. For instance, classical Chinese medicine formulas like “Guizhi Soup” in Zhang Zhongjing’s “Typhoid Theory” feature *Perilla frutescens* as the principal herb for treating fever and headache caused by external wind-cold pathogens. Furthermore, *Perilla frutescens* can also be incorporated with other herbal ingredients to formulate remedies like Scattered Leaves of *Perilla frutescens* or Scattered Stems of *Perilla* that enhance its antiviral properties while promoting surface releasing and dispelling coldness.

This study utilized bioinformatics analysis methods for the first time to identify target genes associated with perilla from multiple databases. The analysis revealed a significant overlap between target genes of *Perilla frutescens* and the genes associated with various viral infections (such as HSV, SARS-CoV-2, influenza virus, and HIV), indicating the substantial antiviral potential of *Perilla frutescens*. GO enrichment analysis and KEGG pathway enrichment analysis conducted using the DAVID platform demonstrated that *Perilla frutescens* primarily participates in biological processes including signal transduction, transcriptional regulation, negative regulation of apoptosis, and inflammatory responses. The results of GO enrichment analysis indicated significant predicted enrichment of target genes of *Perilla frutescens* in biological processes, molecular functions, and cellular components mainly involving regulatory functions and response mechanisms. These analyses provide theoretical support for the antiviral activity of *Perilla frutescens* and serve as a reference for further research on its pharmacological effects and the development of related drugs.

From the literature review, it is evident that *Perilla frutescens* contains key active components such as phenolic compounds and terpenes, which exhibit potent antiviral potential through diverse mechanisms of action against viruses. Among these compounds, certain ones inhibit virus attachment and entry into cells. For instance, thymol prevented HIV-1 entry into target cells by altering the cholesterol content of the viral membrane [44]. Additionally, some compounds interfere with the late stages of virus release. Perillyl alcohol, for example, inhibits the release of infectious HSV-1 particles during maturation in Vero cells [91]. Other compounds like β-sitosterol exert their antiviral activity by directly inactivating viral particles [168]. Furthermore, certain compounds indirectly exert their antiviral effects through immune system regulation. For example, treatment with different doses of Methyl caffeate increased the expression of IL-2, IL-4, IFN-g, soluble Fas in HIV-infected mice [67]. The anti-influenza virus mechanism of pulegone is related to its regulation of IFN-α, IFN-β and IL-2 [46]. Moreover, some compounds demonstrate antiviral activity through antioxidant properties or by inhibiting viral replication protein synthesis or inflammatory response pathways. These compounds exhibit strong antiviral activity without cytotoxicity under tested conditions. In vivo mouse models have also confirmed that these compounds have therapeutic effects on virus-infected mice. Furthermore, combinations of certain compounds show synergistic antiviral effects. For instance, Eugenol combined with acyclovir synergistically inhibited herpes virus replication in vitro [56], while the combination of germacone and oseltamivir demonstrated an additive effect in suppressing influenza virus infection both in vitro and in vivo [101]; these findings provided new insights for developing more effective strategies for antiviral therapy and drug combinations.

However, previous studies have primarily focused on modeling virus infection in vitro using cell lines, with only a limited number of recent studies validating these findings through in vivo experiments in mice. Nevertheless, there is a significant lack of clinical trial data to substantiate the therapeutic effects on humans. Therefore, further clinical studies are necessary to evaluate the safety, efficacy, and potential clinical applications of *Perilla frutescens* as an antiviral agent. Additionally, some active ingredients in *Perilla frutescens* have relatively low bioavailability and do not strictly adhere to Lipinski’s rules [199]. These physicochemical properties may affect drug absorption, distribution, and metabolism thereby impacting their antiviral effects in vivo. Furthermore, although *Perilla frutescens* has demonstrated antiviral activity against a wide range of viruses, it is important to note that different viruses possess unique replication mechanisms and infection routes, leading to the development of diverse diseases. Therefore, it is imperative to conduct an in-depth investigation into the antiviral mechanism of the components derived from *Perilla frutescens*. In addition to the summarized antiviral components above, there are several active components present within *Perilla frutescens* that require further investigation. Future studies should aim at gaining insight into the mechanism of action for different viral infection models using *Perilla frutescens* to gain a more comprehensive understanding of its antiviral activity providing effective strategies for treating virus-related diseases.

In conclusion, *Perilla frutescens* has shown remarkable potential as a potent antiviral agent. Its efficacy in combating viral infections extends beyond humans and encompasses other species as well. Consequently, *Perilla frutescens* holds significant application prospects in the field of antiviral therapy. Given this, it is crucial to further research and develop *Perilla frutescens* and its primary constituents to enhance its antiviral capabilities. Moreover, efforts should be made to mitigate the adverse effects of viral infections on public health by deriving effective prevention strategies from these natural drugs such as *Perilla frutescens*.

## Figures and Tables

**Figure 1 molecules-29-03328-f001:**
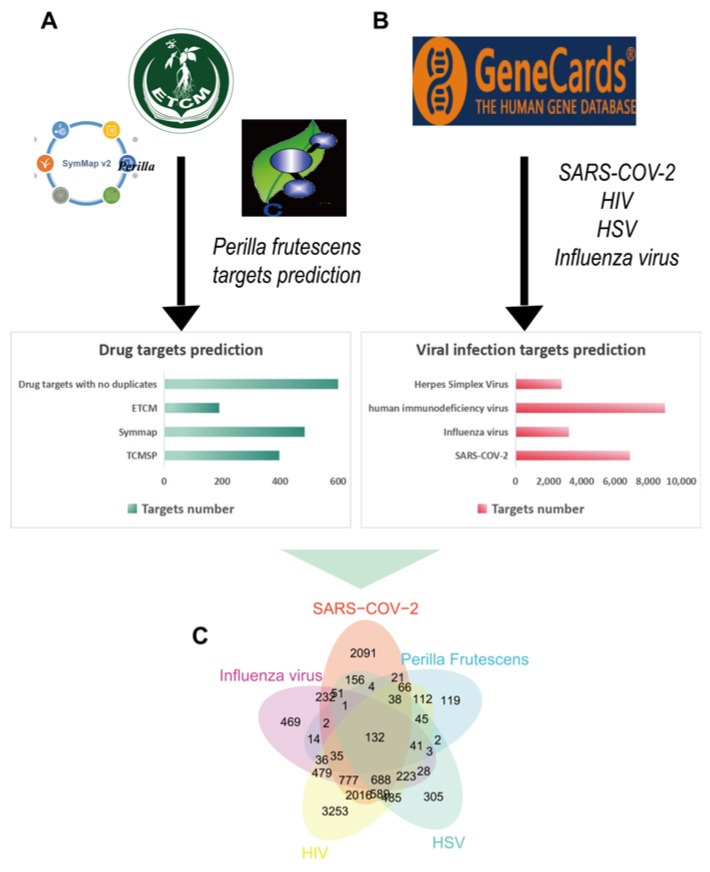
The targets of *Perilla frutescens* against viral infection. (**A**) The number of target genes associated with *Perilla frutescens* retrieved from three publicly available databases. (**B**) The number of target genes associated with prevalent viral pathogens from GeneCards database. (**C**) Venn diagram depicting common target genes between diseases associated with viral infections and *Perilla frutescens*.

**Figure 2 molecules-29-03328-f002:**
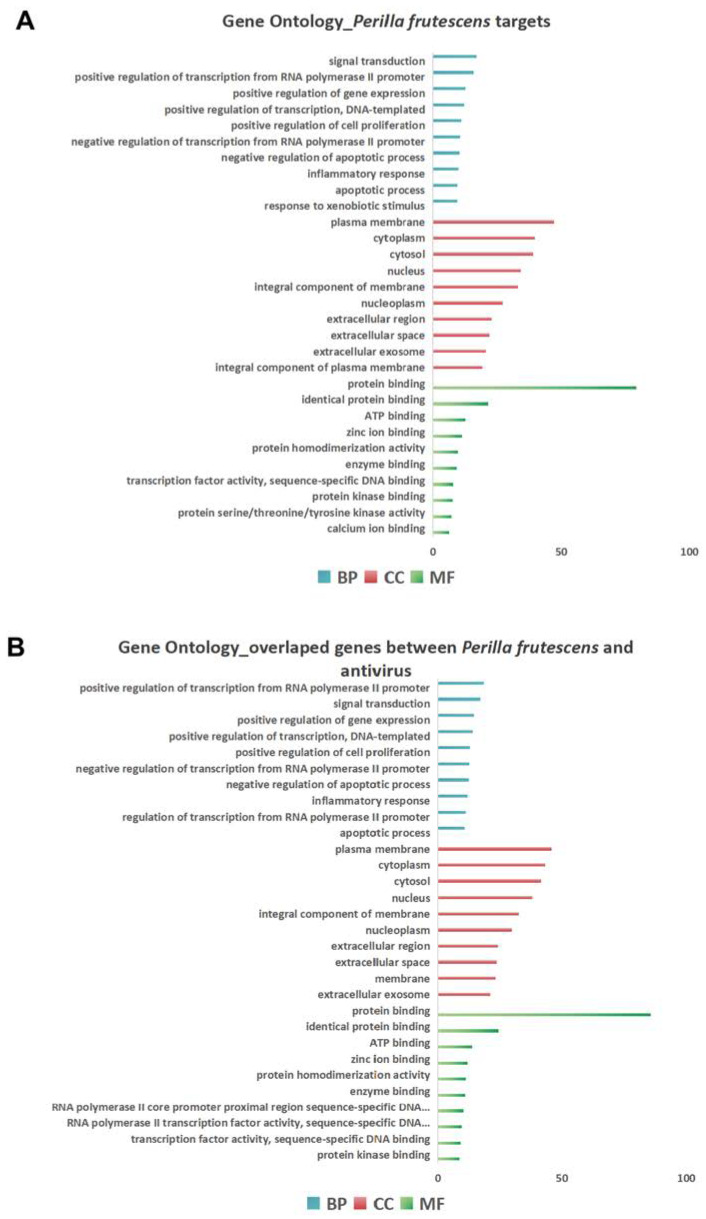
Gene Ontology enrichment analysis revealed the potential of *Perilla frutescens* against viral infection. (**A**) The top 10 results of enrichment analyses for biological processes (BP), cellular components (CC), and molecular functions (MF) were obtained based on the predicted targets of *Perilla frutescens*. (**B**) The top 10 results of enrichment analyses for biological processes (BP), cellular components (CC), and molecular functions (MF) were obtained based on the overlap genes between *Perilla frutescens* and diseases associated with viral infections.

**Figure 3 molecules-29-03328-f003:**
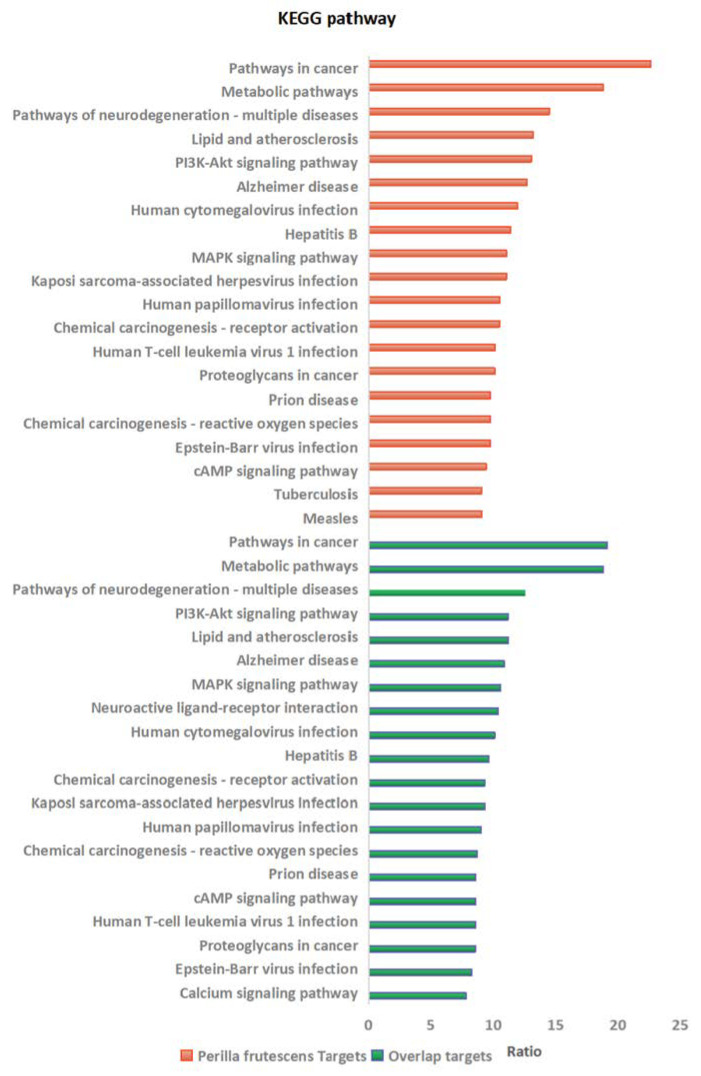
The top 20 enrichments of the KEGG signal pathway associated with the genes of predicted *Perilla frutescens targets* (red) and the overlap genes between drug and disease (green).

**Table 1 molecules-29-03328-t001:** The main phenolic compounds exhibiting antiviral activity in *Perilla frutescens*.

Compound’s Name	Chemical Structure	OB (%)	The Drug’s Efficacy against Specific Virus
Thymol	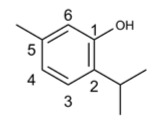	41.47	Influenza virus [45], HSV [43], HIV-1 [44], SARS-CoV-2 [46], FCV [48], MNV [48], BVDV [49], Tomato Leaf Curl New Delhi Virus [50], CyHV-3 [51]
Eugenol	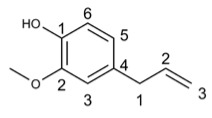	56.24	IAV [54], HSV [55], SARS-CoV-2 [57], TGEV [53], Dengue virus [59], TYLCV [60], TYLCTHV [61], Human Norovirus [62], TMV [63], HCV [63], FCV [64]
Protocatechualdehyde	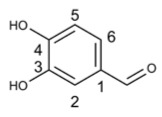	38.35	HBV [65], SARS-CoV-2 [66]
Methyl caffeate	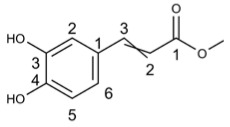	30.68	HIV [67]

**Table 2 molecules-29-03328-t002:** The physicochemical properties of the main phenols in *Perilla frutescens* were assessed using Molinspiration cheminformatics.

Compound’s Name	mi Log P	MW	n-ON	n-OHNH	TPSA
Thymol	3.34	150.22	1	1	20.23
Eugenol	2.10	164.20	2	1	29.46
Protocatechualdehyde	0.76	138.12	3	2	57.53
Methyl caffeate	1.56	194.19	4	2	66.76
Standard criteria	≤5	≤500	≤10	≤5	

**Table 3 molecules-29-03328-t003:** Bioactivity scores of main phenols in *Perilla frutescens* based on Molinspiration cheminformatics.

Compound’s Name	GPCR Ligand	Ion Channel Modulator	Kinase Inhibitor	Nuclear Receptor Ligand	Protease Inhibitor	Enzyme Inhibitor
Thymol	−1.05	−0.53	−1.29	−0.78	−1.34	−0.57
Eugenol	−0.86	−0.36	−1.14	−0.78	−1.29	−0.41
Protocatechualdehyde	−1.25	−0.47	−1.25	−0.88	−1.66	−0.65
Methyl caffeate	−0.62	−0.32	−0.82	−0.26	−0.78	−0.22

**Table 4 molecules-29-03328-t004:** The main terpenoids exhibiting antiviral activity in *Perilla frutescens*.

Compound’s Name	Chemical Structure	OB (%)	The Drug’s Efficacy against Specific Virus
Perillyl alcohol	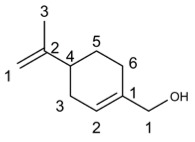	46.24	HSV-1 [91], RSV [95]
Germacron	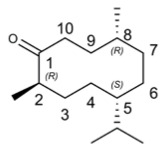	32.50	IAV [100], IVB [101], PPV [97], PRRSV [97], FCV [97], PRV [105]
Patchouli alcohol	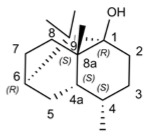	101.96	IAV [17], SARS-CoV-2 [120]
Menthol	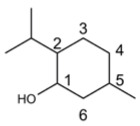	43.31	Coxsackievirus B [124], HSV-1 [125], PrV [125]

**Table 5 molecules-29-03328-t005:** The physicochemical properties of the main terpenoids in *Perilla frutescens* were assessed using Molinspiration cheminformatics.

Compound	mi Log P	MW	n-ON	n-OHNH	TPSA
Perillyl alcohol	2.37	152.24	1	1	20.23
Germacron	4.41	218.34	1	0	17.07
Patchouli alcohol	3.94	222.37	1	1	20.23
Menthol	3.33	156.27	1	1	20.23
Standard criteria	≤5	≤500	≤10	≤5	

**Table 6 molecules-29-03328-t006:** Bioactivity scores of the main terpenoids in *Perilla frutescens* based on Molinspiration cheminformatics.

Compound	GPCR Ligand	Ion Channel Modulator	Kinase Inhibitor	Nuclear Receptor Ligand	Protease Inhibitor	Enzyme Inhibitor
Perillyl alcohol	−0.61	0.04	−1.31	0.03	−0.93	0.14
Germacron	−0.36	−0.14	−1.01	−0.03	−0.66	0.25
Patchouli alcohol	−0.12	0.37	−0.88	0.55	−0.32	0.40
Menthol	−0.76	−0.30	−1.36	−0.60	−0.67	−0.22

**Table 7 molecules-29-03328-t007:** The main flavonoids exhibiting antiviral activity in *Perilla frutescens*.

Compound’s Name	Chemical Structure	OB (%)	The Drug’s Efficacy against Specific Virus
Luteolin	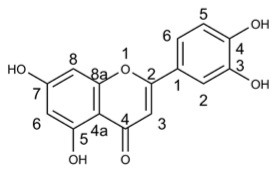	36.16	PRV [128], CHIKV [139], JEV [140], EV71 [142], CA16 [142], FMDV [143], SARS-CoV-2 [132]
Apigenin	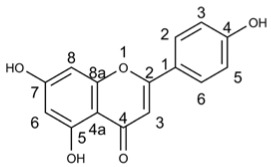	23.06	BPXV [151], IAV [144], FMDV [152], EBV [153], HSV [154], EV71 [155], HCV [157], DENV [159,160]

**Table 8 molecules-29-03328-t008:** The physicochemical properties of the main flavonoids in *Perilla frutescens* were assessed using Molinspiration cheminformatics.

Compound’s Name	mi Log P	MW	n-ON	n-OHNH	TPSA
Luteolin	1.73	286.24	6	4	111.12
Apigenin	2.46	270.24	5	5	90.89
Standard criteria	≤5	≤500	≤10	≤5	

**Table 9 molecules-29-03328-t009:** Bioactivity scores of the main flavonoids in *Perilla frutescens* based on Molinspiration cheminformatics.

Compound’s Name	GPCR Ligand	Ion Channel Modulator	Kinase Inhibitor	Nuclear Receptor Ligand	Protease Inhibitor	Enzyme Inhibitor
Luteolin	−0.02	−0.07	0.26	0.39	−0.22	0.28
Apigenin	−0.07	−0.09	0.18	0.34	−0.25	0.26

**Table 10 molecules-29-03328-t010:** The main sterols exhibiting antiviral activity in *Perilla frutescens*.

Compound’s Name	Chemical Structure	OB (%)	The Drug’s Efficacy against Specific Virus
β-sitosterol	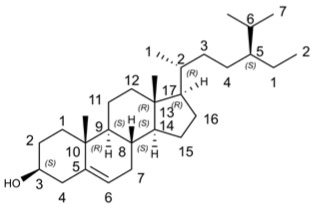	36.91	HIV [170], HBV [170,171], IAV [169], SARS-CoV-2 [165], WSSV [167], HSV-2 [168], TMV [168], Dengue virus-2 [172]

**Table 11 molecules-29-03328-t011:** The physicochemical properties of the main sterols in *Perilla frutescens* were assessed using Molinspiration cheminformatics.

Compound’s Name	mi Log P	MW	n-ON	n-OHNH	TPSA
β-sitosterol	8.62	414.72	1	1	20.23
Standard criteria	≤5	≤500	≤10	≤5	

**Table 12 molecules-29-03328-t012:** Bioactivity scores of the main sterols in *Perilla frutescens* based on Molinspiration cheminformatics.

Compound’s Name	GPCR Ligand	Ion Channel Modulator	Kinase Inhibitor	Nuclear Receptor Ligand	Protease Inhibitor	Enzyme Inhibitor
β-sitosterol	0.14	0.04	−0.51	0.73	0.07	0.51

**Table 13 molecules-29-03328-t013:** The main aldehydes exhibiting antiviral activity in *Perilla frutescens*.

Compound’s Name	Chemical Structure	OB (%)	The Drug’s Efficacy against Specific Virus
Cinnamaldehyde	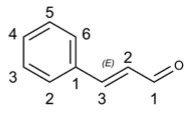	31.99	SARS-CoV-2 [184], IBV [185], influenza A/PR/8 virus [186], CVB3 [187], VMC [187]

**Table 14 molecules-29-03328-t014:** The physicochemical properties of the main aldehydes in *Perilla frutescens* were assessed using Molinspiration cheminformatics.

Compound’s Name	mi Log P	MW	n-ON	n-OHNH	TPSA
Cinnamaldehyde	2.48	132.16	1	0	17.07
Standard criteria	≤5	≤500	≤10	≤5	

**Table 15 molecules-29-03328-t015:** Bioactivity scores of the main aldehydes in *Perilla frutescens* based on Molinspiration cheminformatics.

Compound’s Name	GPCR Ligand	Ion Channel Modulator	Kinase Inhibitor	Nuclear Receptor Ligand	Protease Inhibitor	Enzyme Inhibitor
Cinnamaldehyde	−1.09	−0.39	−1.24	−0.96	−0.79	−0.46

**Table 16 molecules-29-03328-t016:** Other compounds exhibiting antiviral activity in *Perilla frutescens*.

Compound’s Name	Chemical Structure	OB (%)	The Drug’s Efficacy against Specific Virus
Osthol	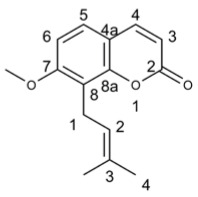	38.75	TMV [189], PCV [190,191]
Piperitenone	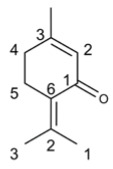	40.05	HSV-1 [192]
Pulegone	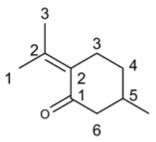	51.6	HSV-1 [193], PrV [193], SARS-CoV-2 [46], influenza virus [46]

**Table 17 molecules-29-03328-t017:** Physicochemical properties of other compounds in *Perilla frutescens* were assessed using Molinspiration cheminformatics.

Compound’s Name	mi Log P	MW	n-ON	n-OHNH	TPSA
Osthol	3.83	244.29	3	0	39.45
Piperitenone	2.56	150.22	1	0	17.07
Pulegone	2.52	152.24	1	0	17.07
Standard criteria	≤5	≤500	≤10	≤5	

**Table 18 molecules-29-03328-t018:** Bioactivity scores of other compounds in *Perilla frutescens* based on Molinspiration cheminformatics.

Compound’s Name	GPCR Ligand	Ion Channel Modulator	Kinase Inhibitor	Nuclear Receptor Ligand	Protease Inhibitor	Enzyme Inhibitor
Osthol	−0.49	−0.59	−0.72	−0.05	−0.77	0.10
Piperitenone	−1.14	−0.84	−1.88	−0.74	−1.25	−0.20
Pulegone	−1.33	−0.81	-2.13	−0.86	−1.09	−0.48

## Data Availability

Not applicable.

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
