# Peer review of "The Antiviral Potential of Perilla frutescens: Advances and Perspectives"

_molecules, 2024, doi:10.3390/molecules29143328_

Round 1

Reviewer 1 Report

Comments and Suggestions for Authors

The article titled: The antiviral potential of Perilla frutescens: a systematic review it is a review that describe in vivo mouse experiments and in vitro cell experiments. This is a very interesting article because it shows all the potential of the Perilla frutescens plant in the treatment of various diseases. In addition, all the monomeric constituents of Perilla Frutescens were evaluated to analyze the potential of each constituent. Thus, the potential of each chemical substance present in the plant was reported. The description of these compounds is very detailed and has several tables summarizing the targets, the physical-chemical properties and bioactivity and to summarize all the antiviral activities.

Overall, the review is very complete and informative, but I believe it will be of interest more to those who want to study the mechanisms of interaction between Perilla frutescens and its various targets.I consider the article suitable for publication in molecules after minor revisions above:

In line 106 = missing ' in:  Perilla frutescens'

In page 4 = increase Figure 2 resolution.

In pages 7, 10, 12 and 13 = increase the resolution of the Tables 1, 4, 7 and 10, respectively.

In page 14 = In table 13, improve the resolution of the chemical structures, and the structure has an error in the drawing between atons 2 and 3.

In page 16 = Table 16 - improve the resolution and standardize the size of the structures presented.

Author Response

The valuable suggestions from the reviewer were greatly appreciated, as his/her input significantly enhanced the quality of the paper and laid a strong foundation for our future research endeavors.

Comments 1: In line 106 = missing ' in:  Perilla frutescens'

Response 1: Thank you for pointing this out, and the sentence has been revised as follows: Furthermore, the GO enrichment analysis for overlapping genes between Perilla frutescens and diseases associated with viral infections aligned with the GO results for total targets of Perilla frutescens as depicted in Fig 2.B. The contents marked in red were incorporated into the original manuscript.

Comments 2: In page 4 = increase Figure 2 resolution.

Response 2: The resolution has been revised to 800dpi, and I appreciate you bringing this to my attention.

Comments 3: In pages 7, 10, 12 and 13 = increase the resolution of the Tables 1, 4, 7 and 10, respectively.

Response 3: Thank you for pointing this out, The resolution has been revised to 800dpi.

Comments 4: In page 14 = In table 13, improve the resolution of the chemical structures, and the structure has an error in the drawing between atons 2 and 3.

Response 4: The resolution has been revised to 800dpi, and the correct structure in table 13 has also been amended. Thank you for bringing this to our attention.

Comments 5: In page 16 = Table 16 - improve the resolution and standardize the size of the structures presented.

Response 5: The resolution has been adjusted to 800dpi, and the structures have been standardized to have the same length throughout.

Reviewer 2 Report

Comments and Suggestions for Authors

I have completed a thorough review of the manuscript titled "The Antiviral Potential of Perilla frutescens: A Systematic Review”. The authors review the potential application of Perilla frutescens as a source for discovering antiviral monomers. The manuscript is written in satisfactory English, but several revisions are needed to enhance its quality and readiness for publication.

·      If the focus is on humans, specify this in the title.

·      The keywords are not representative.

·      In line 27, please remove "microorganism" from the virus definition.

·      Please rewrite lines 40 to 43 to avoid repetition. Additionally, avoid expressions such as “kill virus”.

·      In line 51, explain what "OB" means.

·      In line 52, include a space between “frutescens” and “were”.

·      In lines 69 and 94, "Homo sapiens" must be italicized.

·      In line 99, "david" should be capitalized to "DAVID".

·      There are many examples focused on non-human viruses. Ensure consistency with the statement: “Our focus was specifically on studying Homo sapiens as our selected species”.

·      Define “n-ON”, “n-OHNH”, and “TPSA”. Explain the meaning of standard criteria and why TPSA does not have a standard criterion.

·      Italicize “in vitro” and “in vivo”.

·      In line 271, "influenza A virus (IAV)" was previously defined as "influenza virus A (IVA)".

·      In Table 4, explain how the authors obtained the OB percentage and address the issue of OB percentages exceeding 100% (e.g., Patchouli alcohol).

·      In line 301, clarify why the authors changed the criteria to 30% when line 52 established 30% as a threshold.

·      In line 286, define "NA".

·      In line 321, replace COVID19 with SARS-CoV2.

·      The manuscript needs a deeper discussion on the importance of each parameter of the main compounds specificized in the tables.

·      Osthol is not a potential antiviral compound for human viral infections.

·      The conclusion is poor and needs to be strengthened.

By addressing these points, the manuscript will be significantly improved and better prepared for publication.

Comments on the Quality of English Language

The quality of the English is acceptable. However, it could be improved by using synonyms and different grammatical constructions to avoid making the manuscript seem repetitive.

Author Response

The valuable suggestions from the reviewer were greatly appreciated, as his/her input significantly enhanced the quality of the paper and laid a strong foundation for our future research endeavors.

Comments 1: If the focus is on humans, specify this in the title.

Response 1: Thank you for bringing this to my attention. The review primarily focuses on the antiviral properties of Perilla frutescens in humans. However, certain monomers also demonstrated potential for antiviral activity against other species. For instance, Osthol exhibited effective inhibition against Porcine circovirus type 2 and tobacco mosaic virus infections. After considering the fact that certain pathogens can infect both humans and animals, we have also gathered lectures reporting on the antiviral activity of monomers found in Perilla frutescens. This will provide additional references for various research fields in antiviral studies. However, your suggestion has prompted us to carefully reconsider the manuscript's title, resulting in a modification that better aligns with our research: The antiviral potential of Perilla frutescens: Advances and Perspectives

The contents marked in red were incorporated into the original manuscript.

Comments 2: The keywords are not representative.

Response 2: Thank you for your careful revision and pointing out our oversight, the keywords have been modified as follows: Perilla frutescens, antiviral, natural drug, active compounds, oral bioavailability(OB).

The contents marked in red were incorporated into the original manuscript.

Comments 3: In line 27, please remove "microorganism" from the virus definition.

Response 3: Thank you for pointing this, and the sentence was revised as follows: The role of viral infection in human diseases is significant, and ensuring the prevention of viral infection is a crucial aspect in safeguarding public health. 

The contents marked in red were incorporated into the original manuscript.

Comments 4: Please rewrite lines 40 to 43 to avoid repetition. Additionally, avoid expressions such as “kill virus”.

Response 4: Thank you for pointing this, the contents has been revised as follows: The antiviral potential of numerous natural compounds has been demonstrated in various studies, revealing the ability of numerous plant extracts and secondary metabolites to effectively inhibit viral replication and transmission [14]. The mechanisms of antiviral action are diverse, encompassing interference with viral entry into host cells, inhibition of viral gene expression, disruption of viral assembly, and augmentation of the host immune response [15]. For instance, flavonoids primarily inhibit viral protease activity to prevent viral replication [16]. On the other hand, terpenoids mainly interfere with the fusion of viruses and host cell membranes to impede virus entry into host cells [17]. Additionally, certain polyphenolic compounds directly hinder the cytopathic effect [18]. These findings establish a crucial scientific foundation for the development of novel antiviral medications.

The contents marked in red were incorporated into the original manuscript.

Comments 5: In line 51, explain what "OB" means.

Response 5: Thank you for pointing this, the explain of it was added in the original manuscript as follows: ......was used to sort all reported monomer components of Perilla frutescens by oral bioavailability(OB). The term "oral bioavailability" (OB) refers to the extent and rate at which a drug is absorbed into the systemic circulation. It serves as a crucial parameter for objectively assessing both the oral bioavailability and intrinsic quality of a drug, while also serving as a pivotal criterion for determining its potential as a therapeutic agent. A higher OB value indicates an increased likelihood of clinical development for the compound[26].

The contents marked in red were incorporated into the original manuscript.

Comments 6: In line 52, include a space between “frutescens” and “were”.

Response 6: The revision you provided was greatly appreciated, and the modifications have been made accordingly.

Comments 7: In lines 69 and 94, "Homo sapiens" must be italicized.

Response 7: Thank you for pointing this and the modification has been revised accordingly.

Comments 8: In line 99, "david" should be capitalized to "DAVID".

Response 8: Thank you for pointing this and the modification has been revised accordingly.

Comments9: There are many examples focused on non-human viruses. Ensure consistency with the statement: “Our focus was specifically on studying Homo sapiens as our selected species”.

Response 9: Thank you for pointing this and the sentence has been revised as follows: To further investigate the biological function of Perilla frutescens, we employed the Database for Annotation, Visualization and Integrated Discovery (DAVID, https://david.ncifcrf.gov/) to analyze the complete set of target genes associated with Perilla frutescens, and conducted separate analyses for Gene Ontology(GO) enrichment and Kyoto Encyclopedia of Genes and Genomes(KEGG) pathway enrichment of these target genes.

The contents marked in red were incorporated into the original manuscript.

Comments 10: Define “n-ON”, “n-OHNH”, and “TPSA”. Explain the meaning of standard criteria and why TPSA does not have a standard criterion.

Response 10: Thank you for pointing this and the modification has been revised accordingly: 

The evaluation of drug similarity is crucial in the production and upgrading of drug entities [68]. We first predicted the physicochemical properties of main phenols according to Lipinski’s rule of five(Ro5) using Molinspiration cheminformatics(https://molinspiration.com/). The criteria for the Rule of Five (Ro5) are as follows: LogP should be less than or equal to 5, molecular weight (MW) should be less than or equal to 500 Da, the number of hydrogen bond acceptors (n-ON) should be less than or equal to 10, and the number of violations in terms of hydrogen bond donors (n-OHNH) should be less than or equal to 5. The compounds that conform to the Rule of Five (Ro5) exhibit improved pharmacokinetic properties, enhanced bioavailability in biological metabolism, and therefore possess a higher likelihood of being developed into oral medications [69]. The topological polar surface area (TPSA), which utilizes functional group contributions derived from a comprehensive database of structures, serves as a convenient metric for quantifying the extent of polar surface area [70] and TPSA value ≤140 Å represents good oral bioavailability [68].

The contents marked in red were incorporated into the original manuscript.

Comments 11: Italicize “in vitro” and “in vivo”.

Response 11: Thank you for pointing this and all the font of two proprietary terms in this manuscript were revised accordingly.

Comments 12: In line 271, "influenza A virus (IAV)" was previously defined as "influenza virus A (IVA)".

Response 12: Thank you for your careful revision. However, based on the NCBI-Taxonomy, ICTV-Taxonomy, and the latest research paper on this virus, it is more appropriate to refer to the influenza A virus (IAV). Here are the references and links for your review.

  1. Ho JSY, Angel M, Ma Y, et al. Hybrid Gene Origination Creates Human-Virus Chimeric Proteins during Infection. Cell. 2020;181(7):1502-1517.e23. doi:10.1016/j.cell.2020.05.035
  2. Momont C, Dang HV, Zatta F, et al. A pan-influenza antibody inhibiting neuraminidase via receptor mimicry [published correction appears in Nature. 2023 Jul;619(7970):E50. doi: 10.1038/s41586-023-06385-x]. Nature. 2023;618(7965):590-597. doi:10.1038/s41586-023-06136-y
  3. https://www.ncbi.nlm.nih.gov/Taxonomy/Browser/wwwtax.cgi?id=11320

Comments 13: In Table 4, explain how the authors obtained the OB percentage and address the issue of OB percentages exceeding 100% (e.g., Patchouli alcohol).

Response 13: Thank you for your careful revision. Despite the fact that TCMSP data indicated an OB greater than 100% for this compound, We conducted a comprehensive literature review and identified the following physiological mechanism that could potentially elucidate such elevated oral bioavailability (OB): For instance, biliary excretion into the intestines is subsequently reabsorbed back into the systemic circulation. Moreover, it should be noted that the OB calculation formula() encompasses various factors including test point selection, linear fit degree, et al. In summary, a higher OB value signifies enhanced drug absorption and improved systemic availability. The aforementioned reasons only represent a fraction of the numerous factors, yet they still indicate a significant probability of OB exceeding 100%. Here are the references for you review.

  1. Baghdasaryan A, Fuchs CD, Österreicher CH, et al. Inhibition of intestinal bile acid absorption improves cholestatic liver and bile duct injury in a mouse model of sclerosing cholangitis.J Hepatol. 2016;64(3):674-681. doi:10.1016/j.jhep.2015.10.024
  2. Takahashi K, Ogra Y. Identification of the biliary selenium metabolite and the biological significance of selenium enterohepatic circulation. 2020;12(2):241-248. doi:10.1039/c9mt00274j
  3. Benet LZ, Zia-Amirhosseini P. Basic principles of pharmacokinetics. Toxicol Pathol. 1995 Mar-Apr;23(2):115-23. doi: 10.1177/019262339502300203.

Comments 14: In line 301, clarify why the authors changed the criteria to 30% when line 52 established 30% as a threshold.

Response 14: Thank you for your careful revision. The criteria in the article has been standardized to 20%.

The contents marked in red were incorporated into the original manuscript.

Comments 15: In line 286, define "NA".

Response 15:  Thank you for your careful revision. “NA” has been revised into “neuraminidase”.

The contents marked in red were incorporated into the original manuscript.

Comments 16: In line 321, replace COVID19 with SARS-CoV2.

Response 16: Thank you for pointing this and the modification has been revised accordingly.

Comments 17: The manuscript needs a deeper discussion on the importance of each parameter of the main compounds specificized in the tables.

Response 17: Thank you for pointing this and the modification has been revised as follows: 

The evaluation of drug similarity is crucial in the production and upgrading of drug entities [68]. We first predicted the physicochemical properties of main phenols according to Lipinski’s rule of five(Ro5) using Molinspiration cheminformatics(https://molinspiration.com/). The criteria for the Rule of Five (Ro5) are as follows: LogP should be less than or equal to 5, molecular weight (MW) should be less than or equal to 500 Da, the number of hydrogen bond acceptors (n-ON) should be less than or equal to 10, and the number of violations in terms of hydrogen bond donors (n-OHNH) should be less than or equal to 5. The compounds that conform to the Rule of Five (Ro5) exhibit improved pharmacokinetic properties, enhanced bioavailability in biological metabolism, and therefore possess a higher likelihood of being developed into oral medications [69]. The topological polar surface area (TPSA), which utilizes functional group contributions derived from a comprehensive database of structures, serves as a convenient metric for quantifying the extent of polar surface area [70] and TPSA value ≤140 Å represents good oral bioavailability [68]. The results demonstrated that the main phenols fulfilled the criteria of Rule of Five (Ro5) and exhibited a TPSA value ≤140 Å, as presented in Table 2. The biological activity analysis of compounds, conducted using Molinspiration cheminformatics, encompassed G-protein coupled receptor (GPCR) ligands, ion channel modulators, kinase inhibitors, nuclear receptor ligands, protease inhibitors, and enzyme inhibitors. A bioactivity score >0 indicated promising activity; a score between -0.50 and 0.00 represented moderate activity; while a score ≤-0.50 indicated no activity [68,71]. These findings suggest that phenolic compounds possess moderate affinity as ion channel modulators.

The contents marked in red were incorporated into the original manuscript.

Comments 18: Osthol is not a potential antiviral compound for human viral infections. 

Response 18: Thank you for bringing this to our attention. As mentioned in response to comments 1, we have also gathered lectures on the antiviral activity of monomers found in Perilla frutescens, considering the fact that certain pathogens can infect both humans and animals. These lectures will provide additional references for various research fields in antiviral studies. I hope the reviewer can understand our perspective.

Comments 19: The conclusion is poor and needs to be strengthened.

Response 19: Thank you for your kindly revision, and the modification was revised as follows: In conclusion, Perilla frutescens has demonstrated remarkable potential as a potent antiviral agent. Its efficacy in combating viral infections extends beyond humans and encompasses other species as well. Consequently, Perilla frutescens holds significant application prospects in the field of antiviral therapy. Given this, it is crucial to conduct further research and development on Perilla frutescens and its primary constituents with the objective of enhancing its antiviral capabilities. Moreover, efforts should be made to mitigate the adverse effects of viral infections on public health by deriving effective prevention strategies from these natural drugs such as Perilla frutescens.

The contents marked in red were incorporated into the original manuscript.

Reviewer 3 Report

Comments and Suggestions for Authors

In this manuscript, the author summarized active agents from Perilla frutescens for its potential antiviral properties through bioinformatics analysis. Initially, databases like TCMSP and GeneCards were used to identify related target genes associated with Perilla frutescens and viruses such as SARS-CoV-2, influenza, and HIV. Followed overlapping of target genes between Perilla frutescens and these viruses produced important antiviral potential. Further analysis using the DAVID platform highlighted Perilla frutescens's role in regulating signal transduction, transcriptional processes, apoptosis, and inflammation, crucial for enhancing its antiviral effects. Then all potential antiviral monoers are classified based on cheminformatics tools. However, I would recommend that a major revised manuscript may become acceptable for publication in Molecules after addressing and comments outlined below.

1. The manuscript is bad in scientific writing. Many sentences are difficult to understand and not clear. For example, Line 43-44: “so that patients can further kill virus or inhibit viral replication……”; Line 58-59: “The findings from this review can be utilized to optimize the chemical structure and drug delivery system……” This article neither discusses structural modifications nor addresses drug delivery.

So, I strongly recommend the author check/revise/improve scientific writing of the whole paper to make it clearer and more precise, especially in introduction (part 1), method description (part 2) and discussion (part 4).

2. if possible, please specify the list of target gene after overlapping between Perilla frutescens and virus target gene?  The type of target gene could be specified (directed or indirected) as well, full list of related target gene could be attached in supporting information if possible.

3. Abbreviation showed up in the paper should specified for the first time, i.e. Line 51, OB (oral bioavailability?), Line 95 KEGG, GO etc . 

4. All tables containing chemical structures should be redrew (size, format…. should be aligned) and the stereochemistry of all chiral agents should be clearly specified (Table 1, 4, 7, 10, 13, 16).

Comments on the Quality of English Language

There are lots of typos and spellings, author need to check the whole paper carefully, for example, Line 27, ‘The virus’ should be ‘Viruses’, Line 53, ‘as well as ’ should be ‘using’

Line 42-43,  "so that patients can further kill virus or inhibit viral replication..." could be "resulting in suppression of viral replication and enhancing overall immunity"

Author Response

The valuable suggestions from the reviewer were greatly appreciated, as his/her input significantly enhanced the quality of the paper and laid a strong foundation for our future research endeavors.

Comments 1. The manuscript is bad in scientific writing. Many sentences are difficult to understand and not clear. For example, Line 43-44: “so that patients can further kill virus or inhibit viral replication……”; Line 58-59: “The findings from this review can be utilized to optimize the chemical structure and drug delivery system……” This article neither discusses structural modifications nor addresses drug delivery.

So, I strongly recommend the author check/revise/improve scientific writing of the whole paper to make it clearer and more precise, especially in introduction (part 1), method description (part 2) and discussion (part 4).

Response 1: Thank you for your kind revision. The relevant consents have been appropriately revised in the manuscript. Additionally, all authors thoroughly reviewed the entire manuscript to enhance its language quality using YouDao Dictionary's professional proofreading function, particularly focusing on parts 1, 2, and 4. Specifically, the consent section of part 4 has been rewritten. The syntax modifications were highlighted in purple, while the content modifications were highlighted in red.

Comments 2: if possible, please specify the list of target gene after overlapping between Perilla frutescens and virus target gene?  The type of target gene could be specified (directed or indirected) as well, full list of related target gene could be attached in supporting information if possible.

Response 2: Thank you for your kind revision and the modification was revised accordingly and uploaded with the revised manuscript together.

Comments 3: Abbreviation showed up in the paper should specified for the first time, i.e. Line 51, OB (oral bioavailability?), Line 95 KEGG, GO etc . 

Response 3: Thank you for your kind revision and the modification was revised accordingly.

Comments4: All tables containing chemical structures should be redrew (size, format…. should be aligned) and the stereochemistry of all chiral agents should be clearly specified (Table 1, 4, 7, 10, 13, 16).

Response 4: Thank you for your kind revision. The chemical structures have all been redrawn and utilized using an 800 dpi image, while the stereochemistry of chiral agents has been denoted with an asterisk (*) in the manuscript.

Comments 5: There are lots of typos and spellings, author need to check the whole paper carefully, for example, Line 27, ‘The virus’ should be ‘Viruses’, Line 53, ‘as well as ’ should be ‘using’ Line 42-43,  "so that patients can further kill virus or inhibit viral replication..." could be "resulting in suppression of viral replication and enhancing overall immunity"

Response 5: Thank you for your kind revision and the modification was revised accordingly. Additionally, all authors thoroughly reviewed the entire manuscript to enhance its language quality using YouDao Dictionary's professional proofreading function, particularly focusing on parts 1, 2, and 4. Specifically, the consent section of part 4 has been rewritten.

Reviewer 4 Report

Comments and Suggestions for Authors

In the manuscript “The antiviral potential of Perilla frutescens: a systematic review” The authors summarized the antiviral activity ‘Perilla frutescens” such as bioinformatics analysis, viruses-related genes correlation and signaling pathways. They also summarized its active ingredients and their mechanisms. The manuscript is generally well-addressed and well-cited; however, I have some comments/suggestions.

Line 27: the sentence “the virus are prominent pathogenic microorganisms responsible for causing human........" please, rewrite. Also, this paragraph is general and needs to be revised. It could be broad info. but you could add few examples.

Line 36: please add space before references in text, for example, nasopharyngeal carcinoma[7,8], add space between carcinoma and [7,8]. Revise this all over the manuscript please.

Line 40 : " Many natural Chinese medicines have been reported to have antiviral effects. In addition to the monomer structure that interacts with the virus structure[14]," I suggest adding few examples for these medicines and Perilla frutescens could be one of them that will be detailed in the next paragraph and also add references please so, it will be more specific.

Line 83: in Fig 1,  HSV is included while it is not included in text. Please revise to be consistent.

Line 96: The GO terms ......" please clarify abbreviation of GO (Gene Oncology).

Line 133: " 3.1. Phenols, Polyphenols, also known as phenolic compounds, are aromatic ring-containing ...." This paragraph includes well-known generic info. about phenols, please revise and be specific as possible.

 Line 191: Protocatechualdehyde….Is it effective only with SARS-CoV-2 or with other viruses too? if so, I suggest adding one more example.

 Line 214: Table 2 and Table 3, suppose to add references of these data? Or these are your data based on Molinspiration cheminformatics, please clarify to avoid confusion. Same in other tables.

Line 551: the sentence: “However, the current study was primarily limited to in vivo mouse experiments and in vitro cell experiments, lacking sufficient clinical trial …..” This is confusing. Please revise, did you perform any in-vivo experiments in the current manuscript?

Is Perilla frutescens only cultivated in China so, why called here a Chinese medicine? may add a few lines about this.

References:

References # 2, 5, 19, 30. 31, 37, 45, 61, 66 77, 78, 87, 88, 165, 166, 189, are incomplete. Please revise and follow the journal guidelines.

Comments on the Quality of English Language

Some minor editing of English are required.

Author Response

The valuable suggestions from the reviewer were greatly appreciated, as his/her input significantly enhanced the quality of the paper and laid a strong foundation for our future research endeavors.

Comments1: Line 27: the sentence “the virus are prominent pathogenic microorganisms responsible for causing human........" please, rewrite. Also, this paragraph is general and needs to be revised. It could be broad info. but you could add few examples.

Response 1:Thank you for pointing this for us and the relative content was revised as follows:

The role of viral infection in human diseases is significant, and ensuring the prevention of viral infection is a crucial aspect in safeguarding public health. Certain infectious diseases exhibit extensive spread and high infectivity, thereby influencing the global economy and politics. Furthermore, some viruses can induce chronic infectious conditions such as human immunodeficiency virus (HIV) [1,2], hepatitis B virus (HBV) [3,4], and hepatitis C virus (HCV) [5,6]. These conditions progress gradually and chronically, leading to reduced labor capacity in patients and diminished life expectancy. Consequently, these viruses profoundly affect both the quality of life for patients as well as economic aspects. The association between certain tumors and viruses is well-established, such as Epstein Barr virus with nasopharyngeal carcinoma [7,8], human papillomavirus with cervical cancer [9-11], and human herpesvirus type 8(HHV-8) with Kaposi's sarcoma [12,13]. Viruses exhibit a high mutation rate and continuously generate new variants, posing a significant threat to human health.   

Comments 2: Line 36: please add space before references in text, for example, nasopharyngeal carcinoma[7,8], add space between carcinoma and [7,8]. Revise this all over the manuscript please.

Response 2:Thank you for pointing this for us and the modification was revised accordingly.

Comments 3: Line 40 : " Many natural Chinese medicines have been reported to have antiviral effects. In addition to the monomer structure that interacts with the virus structure[14]," I suggest adding few examples for these medicines and Perilla frutescens could be one of them that will be detailed in the next paragraph and also add references please so, it will be more specific.

Response 3:Thank you for pointing this for us and the modification was revised as follows:

The antiviral potential of numerous natural compounds has been demonstrated in various studies, revealing the ability of numerous plant extracts and secondary metabolites to effectively inhibit viral replication and transmission [14]. The mechanisms of antiviral action are diverse, encompassing interference with viral entry into host cells, inhibition of viral gene expression, disruption of viral assembly, and augmentation of the host immune response [15]. For instance, flavonoids primarily inhibit viral protease activity to prevent viral replication [16]. On the other hand, terpenoids mainly interfere with the fusion of viruses and host cell membranes to impede virus entry into host cells [17]. Additionally, certain polyphenolic compounds directly hinder the cytopathic effect [18]. These findings establish a crucial scientific foundation for the development of novel antiviral medications.

Comments 4: Line 83: in Fig 1,  HSV is included while it is not included in text. Please revise to be consistent.

Response 4:Thank you for pointing this for us and the modification was revised accordingly.

Comments 5: Line 96: The GO terms ......" please clarify abbreviation of GO (Gene Oncology).

Response 5:Thank you for pointing this for us and the modification was revised accordingly.

Comments 6: Line 133: " 3.1. Phenols, Polyphenols, also known as phenolic compounds, are aromatic ring-containing ...." This paragraph includes well-known generic info. about phenols, please revise and be specific as possible.

Response 6: Thank you for pointing this for us and the modification was revised as follows:

Phenols and phenolic ethers are significant scaffolds recurring both in nature and among approved small-molecule pharmaceuticals [28].

Comments 7: Line 191: Protocatechualdehyde….Is it effective only with SARS-CoV-2 or with other viruses too? if so, I suggest adding one more example.

Response 7: Thank you for pointing this. The manuscript has been enhanced by incorporating the anti-HBV activity and relevant reference through carefully literature review.

Comments 8: Line 214: Table 2 and Table 3, suppose to add references of these data? Or these are your data based on Molinspiration cheminformatics, please clarify to avoid confusion. Same in other tables.

Response 8: Thank you for pointing this. The data presented in Table 2, 3, 5, 6, 8, 9, 11, 12, 14, 15, 17 and 18 are exclusively derived from Molinspiration cheminformatics rather than referenced sources. These specific data points have been emphasized in the table title.

Comments 9: Line 551: the sentence: “However, the current study was primarily limited to in vivo mouse experiments and in vitro cell experiments, lacking sufficient clinical trial …..” This is confusing. Please revise, did you perform any in-vivo experiments in the current manuscript?

Response 9: Thank you for pointing this and the sentence has been modified to as follows: However, previous studies have primarily focused on modeling virus infection in vitro cell lines, with only a limited number of recent studies validating these findings through in vivo mouse experiments. Nevertheless, there is a significant dearth of clinical trial data to substantiate the therapeutic effects on humans.

Comments 10: Is Perilla frutescens only cultivated in China so, why called here a Chinese medicine? may add a few lines about this.

Respoonse 10: Thank you for pointing this and the modification has been revised as follows:

The annual herb Perilla frutescens (L.) Britt., belonging to the Labiatae family, exhibits medicinal and culinary properties in traditional Chinese medicine (TCM) [19]. Its dried stems, leaves, and seeds have been utilized as medicinal materials. Perilla frutescens has demonstrated pharmacological activities, ......

Commens 11: References:

References # 2, 5, 19, 30. 31, 37, 45, 61, 66 77, 78, 87, 88, 165, 166, 189, are incomplete. Please revise and follow the journal guidelines.

Response 11: Thank you for pointing this and the modification has been revised accordingly.

Round 2

Reviewer 1 Report

Comments and Suggestions for Authors

The title of the article has improved, as several written parts of the article. However, all the Figures need to improve their resolution, as I stated in my first report.

Author Response

The valuable suggestions from the reviewer were greatly appreciated, as his/her input significantly enhanced the quality of the paper and laid a strong foundation for our future research endeavors.

Comment: The title of the article has improved, as several written parts of the article. However, all the Figures need to improve their resolution, as I stated in my first report.

Response: Thank you for your assistance with our work, all the figures in the manuscript have been changed to 1600dpi resolution. Thank you very much for your careful guidance. 

Reviewer 2 Report

Comments and Suggestions for Authors

Thank you for the clarifications regarding the manuscript. Response 13 was very clear. The quality of the manuscript has now significantly improved. I have only minor considerations:

Regarding comment 12, what I meant is that in the manuscript, both forms are used (e.g., in the second paragraph of the Thymol section, line 172 of the revised version, it states "Influenza Virus A (IVA)," and in the second paragraph of the Patchouli alcohol section, line 312, it states "influenza A virus (IAV)").

Additionally, it might be better to set the threshold at 20% instead of 30%, since apigenin has a bioavailability of 23.06%.

Author Response

The valuable suggestions from the reviewer were greatly appreciated, as his/her input significantly enhanced the quality of the paper and laid a strong foundation for our future research endeavors.

Comments 1: Regarding comment 12, what I meant is that in the manuscript, both forms are used (e.g., in the second paragraph of the Thymol section, line 172 of the revised version, it states "Influenza Virus A (IVA)," and in the second paragraph of the Patchouli alcohol section, line 312, it states "influenza A virus (IAV)").

Response 1: Thank you for your feedback on comment 12. I have reviewed the manuscript and made the necessary revisions. All instances have now been standardized to "IAV" and highlighted in yellow.

Comments 2: Additionally, it might be better to set the threshold at 20% instead of 30%, since apigenin has a bioavailability of 23.06%.

Response 2: Thank you for your suggestion. I have made the necessary adjustments, and the threshold is now set at 20% instead of 30%, considering that apigenin has a bioavailability of 23.06%.

Reviewer 3 Report

Comments and Suggestions for Authors

This manuscript still contains many errors in the table containing chemical structures, eg. Table 1, 7, 10, 13, 16, some of chiral centers are assigned incorrectly, moreover the size and font of structure are not the same. I suggest the author ask a organic chemist for assistance if necessary.  

Author Response

The valuable suggestions from the reviewer were greatly appreciated, as his/her input significantly enhanced the quality of the paper and laid a strong foundation for our future research endeavors.

Comment: This manuscript still contains many errors in the table containing chemical structures, eg. Table 1, 7, 10, 13, 16, some of chiral centers are assigned incorrectly, moreover the size and font of structure are not the same. I suggest the author ask a organic chemist for assistance if necessary.  

Response: Thank you for your valuable feedback on our manuscript. We have redrawn all tables containing chemical structures according to your requirements, setting the font size to 6 and ensuring a resolution of 1440 dpi. We have also explicitly specified the stereochemistry of all chiral centers to ensure precision and accuracy. In response to your valuable suggestion, we have invited Dr. Mengzhu Xue, a drug chemistry research specialist from Shanghai JiaoTong University School of Medicine, to assist us in modifying the drug structure. Furthermore, we would like to express our sincere gratitude by acknowledging her invaluable assistance. Additionally, we have extended authorship to include Associate Professor Haoran Wang, who possesses extensive academic experience at Massey University (New Zealand) and exceptional skills in professional English article writing. We have enlisted her expertise to enhance the quality of our English writing and optimize readability for global readers. 

Please kindly note that the relevant modifications have also been visually emphasized through highlighting in the second round revision.